# The Adjustment of China’s Grain Planting Structure Reduced the Consumption of Cropland and Water Resources

**DOI:** 10.3390/ijerph18147352

**Published:** 2021-07-09

**Authors:** Yu Zhang, Jieyong Wang, Chun Dai

**Affiliations:** 1Institute of Geographic Sciences and Natural Resources Research, Chinese Academy of Sciences, Beijing 100101, China; zhangy1.19b@igsnrr.ac.cn (Y.Z.); daic.19s@igsnrr.ac.cn (C.D.); 2Key Laboratory of Regional Sustainable Development Modeling, Chinese Academy of Sciences, Beijing 100101, China; 3College of Resources and Environment, University of Chinese Academy of Sciences, Beijing 100049, China

**Keywords:** grain production, structure adjustment, cropland and water resources, food security

## Abstract

Driven by technological progress and market demand, the optimization and adjustment of grain planting structure played an important role in increasing grain output. Due to the great difference between the yield per unit area of different types of food crops, the consumption of cropland and water resources has a significant change during the grain growth. From the perspective of structural adjustment, rather than the usual productive factor input, we analyze the process of adjustment for grain planting structure in China and its effect on the consumption of cropland and water resources by using the scenario comparative analysis method. The results show that: (1) From 2003 to 2019, China’s grain output has increased steadily and the planting structure has changed greatly. Rice was replaced by corn to become the grain crop with the maximum proportion of planting area since 2007. The increase of corn planting structure proportion is concentrated in the northern regions. (2) At the national level, according to the adjustment of grain planting structure, the saving of cropland and water resources consumption showed a “cumulative effect” as time went on. (3) The saving effects of structural adjustment in the northern regions on cropland and water resources consumption are better than that in the southern regions, such as Northeast China Plain, Northern arid and semiarid region and Loess Plateau. (4) In reality, although the adjustment of grain planting structure saved lots of cropland and water resources, the continuous growth of grain output has increased the pressure on the ecological environment in the northern regions according to theirs water limits. Therefore, it is necessary to continuously optimize the grain planting structure and restrict land reclamation in northern China. In addition, to ensure food security, it is feasible to encourage the southern regions with abundant water and heat resources to increase the grain planting area and meet its self-sufficiency in grain demand.

## 1. Introduction

Grain production is the main source of cropland and water resource consumption in the world [1]. Statistics also indicate that agricultural water use accounts for about 80% of the global total water consumption [2]. China is an important country for food production and consumption in the world [3]. However, the supply of cropland and water resources is insufficient in China, with per capita occupancy equal to 28% and 40% of the world average, respectively [4,5]. The evolution of China’s grain production pattern is not only reflected in the movement of grain production gravity [6], but also accompanied by the adjustment of grain planting structure [7]. Further, these will cause the problem of spatial allocation of cropland and water resources and ultimately affect the sustainable development of the whole agriculture. Therefore, this is crucial to ensuring China’s food security and promoting agricultural sustainable development, through the analysis of grain planting structure adjustment and its impact on the consumption of cropland and water resources.

With the rapid development of the regional economy and society, great changes have taken place in terms of grain production and supply-demand patterns in China. It has been found the barycenter of grain crops presents a trend of goes to north and center movement in China [6], which is driven by factors such as economic benefits, per capita farmland acreage and grain yield per unit area [8]. In addition, the gravity of China’s grain output is also moving northward due to urbanization, land-use policies and climate change. The pattern of grain supply-demand has changed, from the traditional pattern of “grain in the south being transported to the north” to the present pattern of “grain in the north being transported to the south” [9,10]. Due to 0 °C isotherm moved northward, the annual accumulated temperature (AAT) increased and the rice planting area continued to expand in Northeast China [7,11,12]. The shortage and spatial imbalance of cropland and water resources restrict the sustainable development of grain production. Other studies analyze the utilization of cropland and water resources and their spatial allocation and evaluate the relationship between grain production and the spatial allocation of cropland and water resources [13,14,15].

As one of the most populous countries in the world, China has always paid high attention to food security. Many studies have identified several ways to help increase grain production. We summarize as follows: (1) Expand arable land to increase the area of grain crop planting [16]. (2) Increase the input of productive factors, such as chemical fertilizers, pesticides, etc. [17,18]. (3) Increase the intensity of grain crop planting [19,20]. There are some problems with this approach to increasing food production. On the one hand, the abuse of pesticides and fertilizers causes serious ecological and environmental problems in grain production, such as agricultural non-point source pollution [21], which affects the food quality and further threatens human health. On the other hand, factor input has a marginal effect. That is to say, the increase of production factor input has less and less effect on improving grain output [22]. Moreover, overuse of cultivated land resources leads to the decline of soil fertility, which is not conducive to agricultural sustainable development. In this case, we need to explore more effective strategies to increase grain production [23]. As such, the previous studies focused on the input amount of grain production and crop yield, as well as the driving factors of crop structure adjustment. Few studies have analyzed the effects of grain production on cropland and water resources from the perspective of planting structure. Different grain crops not only have different yields per unit area, but also have great differences in cropland and water consumption. The adjustment of the grain planting structure has become a “bridge” to observe the relationship between grain demand and consumption of cropland and water resources. Therefore, while the total grain yield increases, the adjustment of the internal grain planting structure has an impact on the consumption of cropland and water resources in grain production. In other words, it is very feasible and meaningful to analyze the consumption of cropland and water resources in grain production through adjustment of planting structure rather than the input of production factors. Therefore, this paper abandons the traditional research idea of analyzing grain production based on productive factor input. From the perspective of the adjustment of grain planting structure, to calculate its effects on cropland and water resources consumption while the grain yield continues to increase.

Considering the condition of the shortage and spatial imbalance of cropland and water resources in China, we want to answer two questions: (1) Does the adjustment of the grain planting structure contribute to the economic utilization of cropland and water resources? (2) What are the differences in the consumption of cropland and water resources between different agricultural areas? These questions are posed based on the scenario comparison method, taking the grain planting structure without adjustment as the base scenario in 2003. This paper aims to explore the impact of cropland and water resources by adjusting the grain planting structure from 2003 to 2019. It provides a reference for the rational adjustment of grain production structure and relevant departments to make decisions in the future.

## 2. Materials and Methods

### 2.1. Data Sources and Processing

The agricultural division is an important approach to guide agricultural production [17]. China is divided into nine agricultural regions based on geographical zoning and regional characteristics of the grain planting system, which are Northeast China Plain, Northern arid and semiarid region, Huang-Huai-Hai Plain, Loess Plateau, Qinghai Tibet Plateau, Sichuan Basin and surrounding regions, Middle-lower Yangtze Plain, Yunnan-Guizhou Plateau and Southern China. The study region includes 31 provinces (except Hong Kong, Macao and Taiwan) that participated in the complete information (Figure 1). The basic map of the nine agricultural regions is taken from the Resources and Environmental Sciences and Data Center, Chinese Academy of Science (http://www.resdc.cn/Default.aspx (accessed on 5 March 2021)).

All of the study data are public data obtained from different sources. These data were derived from the following sources. Data sets on grain planting area and yield of rice, wheat, corn, beans and tubers were obtained from the China Statistical Yearbook (2004–2020) (http://www.yearbookchina.com/index.aspx (accessed on 20 March 2021)). Agricultural water consumption data were obtained from the China Water Resources Bulletin of the corresponding years. The basic data of the nine agricultural regions were collected by each province, such as grain output, agricultural water consumption, cropland resources and so on.

Due to the different climatic conditions and soil environment in nine agricultural regions, water resource consumption varies greatly among different crops in the same agricultural region or the same crop in different agricultural regions [24]. Before the water consumption analysis, virtual water content per unit mass of grain crops needs to be considered. Firstly, the weight mean of virtual water content per unit mass of grain crops was calculated in each province by referring to relevant research results. Secondly, the proportions of different grain production were measured in each province. Finally, the virtual water content per unit mass of different grain crops was obtained in nine agricultural regions of China.

### 2.2. Model Design

#### 2.2.1. Calculation Formula of Cropland Resource Consumption

To obtain the same grain yield, the gap between the cropland resources consumption with structural adjustment and without structural adjustment is the number of cropland resources saved by the adjustment of the grain structure. The specific calculation process is shown as follows:(1)$$Q_{t}=A_{t}\times Y_{t}=A_{t}\times \sum \left(s_{it}\times y_{it}\right)$$
(2)$$Q_{t+j}=A_{t+j}\times \sum \left(s_{i,t+j}\times y_{i,t+j}\right);\;j=1,2,3\ldots 16$$
where $Q_{t},Q_{t+j}$ represents the grain yield in *t* and *t + j* period, *j* is the number of the year (in this study, *j* = 16). $A_{t}$ is the sown area of grain, $S_{it}$ represents the sown area of crop *i* accounted for the proportion of the sown total area and $y_{i}$ is the yield per unit area of crop *i*. Similarly, $A_{t+j},\;S_{i,t+j},\;y_{i,t+j}$ represent sown area of grain, sown area of crop *i* accounted for the proportion of the sown total area and the yield per unit area of crop *i* in *t + j* period, respectively.
(3)$$Q_{t+j}=A_{t+j}^{\prime }\times \sum \left(s_{it}\times y_{i,t+j}\right)$$
(4)$$A_{t+j}^{\prime }=A_{t+j}\times \frac{\sum \left(s_{i,t+j}\times y_{i,t+j}\right)}{\sum \left(s_{it}\times y_{i,t+j}\right)}$$

$A_{t+j}^{\prime }$ represents sown area of grain when without structural adjustment. Meanwhile, $S_{it}=S_{i,t+j}$. Therefore, the gap between the cropland resources consumption with structural adjustment and without structural adjustment can be calculated using Equation (5):(5)$$A_{t+j}^{\prime }-A_{t+j}=\left[\frac{\sum \left(s_{i,t+j}\times y_{i,t+j}\right)}{\sum \left(s_{it}\times y_{i,t+j}\right)}-1\right]\times A_{t+j}$$

If $A_{t+j}^{\prime }-A_{t+j}>0$, it means that the adjustment of grain planting structure can help save cropland resources, while if $A_{t+j}^{\prime }-A_{t+j}<0$, it means that the adjustment of grain planting structure increases the consumption of cropland resources and if $A_{t+j}^{\prime }-A_{t+j}=0$, it means there is no effect.

#### 2.2.2. Calculation Formula of Water Resource Consumption

Following the same train of thought, the amount of water resources saving can be calculated by the adjustment of the grain structure. The formula is: (6)$$\beta_{i,t+j}=\frac{\sum \left(\left(s_{i,t+j}\times y_{i,t+j}\right)\right)}{\sum \left(s_{it}\times y_{i,t+j}\right)}$$
(7)$$W_{t+j}^{\prime }-W_{t+j}=A_{t+j}\times \left[\sum \left(s_{it}\times \beta_{i,t+j}\times y_{i,t+j}\times m_{i,t+j}\right)-\sum \left(s_{i,t+j}\times y_{i,t+j}\times m_{i,t+j}\right)\right]$$
where $\beta_{i,t+j}$ is the increase or decrease coefficient, it indicates that the changing intensity is caused by structural adjustment compared with no structural adjustment. In Equation (7), $W_{t+j}^{\prime },W_{t+j}$ represent the amount of water resources consumption with structural adjustment and without structural adjustment, respectively. $m_{i,t+j}$ represents water consumption per unit mass of crop *i*.

If $W_{t+j}^{\prime }-W_{t+j}>0$, it means that the adjustment of grain planting structure contributes to the economic utilization of water resources, while if $W_{t+j}^{\prime }-W_{t+j}<0$, it means that the adjustment of grain planting structure increases the consumption of water resources and if $W_{t+j}^{\prime }-W_{t+j}=0$, it means there is no effect.

#### 2.2.3. Consumption Reduction Contribution (*CRC*) of Cropland and Water Resources

The consumption reduction contribution is used to measure the saving effect of cropland and water resources in an agricultural region. It can provide a concise and intuitive result regardless of the total amount of cropland and water resources among agricultural regions. The formula is
(8)$$CRC_{l}=\frac{A_{t+j}^{\prime }-A_{t+j}}{A_{t+j}^{\prime }}$$
(9)$$CRC_{w}=\frac{W_{t+j}^{\prime }-W_{t+j}}{W_{t+j}^{\prime }}$$

In Equations (8) and (9), $CRC_{l}$ and $CRC_{w}$ represent consumption reduction contribution of cropland resources and consumption reduction contribution of water resources, respectively. The values of $CRC_{l}$ and $CRC_{w}$ range from −1 to 1. The higher the value, the higher the consumption reduction contribution (*CRC*) of cropland and water resources. On the contrary, the smaller the value, the lower the consumption reduction contribution (*CRC*) of cropland and water resources.

The last required a bit of explanation that the study results were calculated by Excel and visualized by ArcGIS and Origin software.

## 3. Results

### 3.1. Changes of Grain Yield and Consumption of Cropland and Water Resources in China

From 2003 to 2019, China’s output of major farm products has shown a steady increase (Figure 2), which can be divided into two stages: the rapid growth stage (2003–2015) and the fluctuating growth stage (2016–2019). During the rapid growth stage (2003–2015), China’s grain output increased from 431 million tons to 621 million tons, with an average annual growth rate of 3.10%. In the fluctuating growth stage (2016–2019), although grain production has declined in some years, it has maintained a rapid growth overall, reaching 664 million tons in 2019.

However, with the rapid increasing of the total grain yield, the consumption of cropland and water resources for grain production remained stable on the whole. The water resource consumption showed a trend of small fluctuation, which remain between 350 × 10^9^ m^3^ and 390 × 10^9^ m^3^. The cropland resource consumption increased from 99.41 × 10^4^ km^2^ in 2003 to 116.06 × 10^4^ km^2^ in 2019, which was only 1.17 times higher. However, grain production increased 1.54 times in 17 years. Therefore, the rapid increase of grain output is not only related to the use of chemical fertilizers and pesticides, technological progress [25] and other input elements, but also closely related to the adjustment of grain planting structure.

### 3.2. Spatio-Temporal Changes of Grain Planting Structure in China

#### 3.2.1. Temporal Changes of Grain Planting Structure in China

We selected five types of grain crops, including rice, wheat, corn, beans and tubers, to analyze the change of grain planting structure in China. Figure 3 shows that significant changes have taken place in grain planting structure during the steady growth stage of grain outputs. Concretely, the proportion of rice, wheat and corn changed from 27.85%, 23.11% and 25.29% in 2003 to 26.30%, 21.01% and 36.56% in 2019, respectively. Since 2007, corn, the maximum proportion of planting structure, has replaced rice as China’s largest grain crop, which has continued to increase.

After China acceded to the World Trade Organization (WTO), a large number of international agricultural products entered the Chinese market, especially the import of soybeans increased sharply. The soybean imports have reached 88.51 million tons in 2019, accounting for 83.10% of China’s total soybean consumption. As a result, China’s grain planting structure has also been strategically adjusted, with the proportion of soybeans planted gradually declining from 13.55% in 2003 to 9.81% in 2019.

#### 3.2.2. Spatial Characteristics of Grain Planting Structure in China

Figure 3 reflects the dynamic change trend of China’s grain planting structure from 2003 to 2019, but does not reflect the spatial distribution characteristics among the nine agricultural regions. Therefore, Using the agricultural division in Figure 1, we show the grain planting structure of different agricultural regions in China at two time points in 2003 and 2019 (Figure 4).

Significant differences in grain planting structure among the nine agricultural regions in China and the proportion of corn planted in the northern region has increased rapidly, which can be presented by Figure 4. For example, the proportion of corn planted in the Northeast China Plain, the Loess Plateau and the Northern arid and semiarid region has increased from 43.21%, 34.74% and 34.76% in 2003 to 55.13%, 55.03% and 55.30% in 2019, respectively. Compared with 2003, in 2019, the planting area of rice continued to expand and the planting proportion increased by 5.8% in the Northeast China Plain, while the planting proportion of beans decreased by 12.43%. Wheat is the dominant crop in Huang-Huai-Hai Plain and the proportion of planting area is stable at about 48%. Rice is the main grain crop in southern China, but its planting proportion decreased in some agricultural regions and was gradually replaced by corn, such as Yunnan-Guizhou, Plateau Sichuan Basin and surrounding regions.

### 3.3. The Impact Analysis of Grain Planting Structure Adjustment on Cropland Resource Consumption

We used Equation (5) to calculate the cropland resource consumption with or without the adjustment of grain planting structure from 2003 to 2019 (Figure 5) to analyze the evolution trend in both scenarios and then used Equation (8) to calculate the consumption reduction contribution (Table 1) to measure the saving effect of cropland resource.

It can be seen from Figure 5 that the cropland resource consumption with structural adjustment is less than that without structural adjustment, indicating that the adjustment of the grain planting structure contributes to saving cropland resources under certain grain output. Note the difference value of the slope of the trend line in both scenarios, it represents the extent to which the structural adjustment affects the cropland resource consumption in this region. From 2003 to 2019, due to the adjustment of grain planting structure, the saving of cropland resources consumption showed a “cumulative effect” at the national total. Consistent with the overall trend of the country, the northern region also showed a significant “cumulative effect”, such as Northeast China Plain and the Northern arid and semiarid region. For the southern region where arable cropland resources are relatively scarce, such as Middle-lower Yangtze Plain, Yunnan-Guizhou Plateau and Southern China, the trend lines of cropland resource consumption under the two scenarios are nearly parallel, indicating that grain planting structural adjustment plays a certain role in the reduction of cropland resources, but the reduction scope is almost to the limit.

At the national total, 5.21 × 10^4^ km^2^ of cropland resources saved by the adjustment of grain planting structure and consumption reduction contribution of cropland resources (CRC_l_) was 4.62%. From the perspective of regions, both the area saved and the CRC_l_ of the northern region were the highest. More concretely, because high-yielding rice and corn replaced the relatively low-yielding wheat and soybean, area saved of cropland resources and CRC_l_ in Northeast China Plain reached 3.62 × 10^4^ km^2^ and 15.61%, respectively, which were higher than other agricultural regions. Followed by the Northern arid and semiarid region, area saved of cropland resources and CRC_l_ were 1.58 × 10^4^ km^2^ and 13.93% respectively, which were mainly caused by the substitution of planting corn for wheat. However, compared with the substantial expansion of the actual grain planting area in northern China, the cropland resources saved by grain planting structural adjustment were still quite limited, accounting for only 29.35% of the actual increase of the grain sown area in the same period.

Different from the northern region, the proportion of high-yielding rice planting is relatively high in the southern region, such as Middle-lower Yangtze Plain and Southern China, so the scope for the adjustment of the grain planting structure becomes very limited. Therefore, the cropland resource area saved by the adjustment of the grain planting structure is less than 1 × 10^4^ km^2^ and the CRC_l_ value is also quite low in each agricultural region. In other words, the adjustment of the grain planting structure has a small effect on saving cropland resources in China’s southern region. However, in reality, except for the Middle-lower Yangtze Plain, the grain planting area of other agricultural regions has shrunk to some extent.

### 3.4. The Impact Analysis of Grain Planting Structure Adjustment on Water Resource Consumption

Figure 6 shows the evolution trend of water resource consumption with or without the adjustment of grain planting structure in nine agricultural regions and national total from 2003 to 2019. At the national level, the adjustment of the grain planting structure has a crucial effect on saving water resources. The gap of water resource consumption under the two scenarios became larger over time in the process of grain production, which also showed a significant “cumulative effect”.

Compared with the southern regions and northern regions, we found that in the northern regions where water resources supply is insufficient, the adjustment of grain planting structure has a more obvious effect on saving water consumption. For example, compared with the non-adjustment scenario, water resource consumption continues to decrease in the Northeast China Plain, Loess Plateau and Northern arid and semiarid region. From the perspective of development trends, it is still possible to continue to save more water resources. However, in water-rich southern regions, such as Middle-lower Yangtze Plain and Southern China, the trend lines of water consumption almost coincide with or without the adjustment of grain planting structure, indicating that the adjustment of grain planting structure has a small effect on water resource reduction and reaches the upper limit.

It can be seen from Table 2 that the grain planting structural adjustment has helped to save an appropriate 437.09 × 10^8^ m^3^ water resource for the country, compared with the scenarios without structure adjustment, accounting for 38.58% of the increased water resource consumption in actual grain production. At the regional level, Qinghai Tibet Plateau has the largest value of consumption reduction contribution of water resources (CRCw)under both scenarios, followed by the Northern arid and semiarid region, Loess Plateau and Northeast China Plain. However, the Qinghai-Tibet Plateau has a low grain planting area and limited water resource consumption, so its reference value is insufficient. Thus, the Northern Region has a greater effect on water resource reduction. In particular, because of the rare rainfall all the year-round, the water resource consumption of grain production mainly depends on irrigation water in the Northern arid and semiarid region [26], which is part of the temperate continental climate region. CRCw is 14.68%, higher than in other agricultural regions. Through grain planting structural adjustment, 102.27 × 10^8^ m^3^ of water resources were saved, accounting for 47.98% of the actual increase in the total water demand for grain production in this region.

The southern region is located in subtropical and tropical climates and has abundant water resources. Statistics also show that the water resources per unit arable area in southern regions is 8 times that in northern regions [27]. The value of water saved and CRCw are relatively low in the southern regions. Middle-lower Yangtze Plain, which is the most important grain planting area in the southern regions, the water resources saved accounted for about 1/30 of the actual increase in water resources consumption due to the expansion of grain planting area. Even, the water resources consumption has further increased by grain planting structural adjustment in Southern China and CRCw is −1.51%.

## 4. Discussion

As the population grows and wealth increases, people can buy more varied and resource-intensive diets. For now, the human demand for food is still increasing. It has been found that global agricultural production may need to increase by 70–110% to meet the growing demands associated with human uses and livestock feed by 2050 [28]. There is increased competition for land, water, energy and other inputs into food production [29]. The expansion of built-up areas occupies a lot of arable lands in the process of rapid urbanization, crowding out the space for grain production [30]. Recent studies also show that future urbanization will result in a 1.8–2.4% loss of global cultivated lands by 2030 [31,32]. The sudden outbreak of COVID-19 has alerted the world to the importance and seriousness of food security [33,34]. Global food security faces even more severe challenges in the future. However, China has achieved a sustained and steady increase in grain output in the process of rapid urbanization, which has ensured the country’s food security. The study results prove that the proportion of grain planting increased, such as corn, which alleviated the pressure of grain production on the consumption of cropland and water resources to a certain extent.

Remarkably, with the improvement of technology and living standard, people pay more attention to the nutritional content in food consumption, which further enhances the substitutability among different food crops demands. For the past few years, the consumption quantity of animal food increases year by year, especially the consumption of milk and dairy product. At the same time, grain consumption decreases significantly. This reflects the replacement of basic rations with feed and industrial grains. In China, the food structure of residents is transforming from vegetable fiber orientation to animal fat and high protein orientation. FAO statistics on the food supply of countries and regions indicate that per capita nutritional level of the Chinese Mainland is close to Japan, Taiwan and Korea, which is 3 kcal each day [35,36]. Consumer demand tends to be more about the nutrient content of a grain crop than its type. Therefore, the structure of grain production weakens the constraints on food supply. In that sense, the adjustment of the grain planting structure becomes an inevitable trend. It is possible to increase grain production and reduce the pressure of cropland and water resource consumption by increasing the planting area of high-yielding crops in the context of marketization.

However, the study also found that the adjustment of grain planting structure of the corn and other crops were concentrated in the northern regions, such as the Northern arid and semiarid region, the Northeast China Plain and the Loess Plateau, which are also areas with severe water shortage. Although the structural adjustment has improved the utilization efficiency of cropland and water resources, the growing acreage of food crops has further aggravated the contradiction of water shortage in northern regions. It has brought more pressure to the ecologically fragile northern regions. By contrast, the southern regions with superior water and heat conditions are suitable for grain production. The loss of the agricultural labor force has resulted in the abandonment of arable land [37,38], which threatens agricultural production and food security. In a word, the natural conditions for grain production in the northern regions, such as heat and water resources, are inferior to those in the southern regions. The spatial movement of grain production barycenter and the spatial dynamic change of grain planting structure aggravated the spatial imbalance of cropland and water resources to some extent. What should be paid attention to is the spatial dislocation between the expansion of grain production and the natural resources, such as cropland, heat and water resources.

Therefore, future research should focus on the following points: First, the regional types of grain production in China are classified, such as core area, potential area and buffer area. The purpose is to clarify the ability to ensure self-sufficiency in food demands. Second, the trend of grain planting structure adjustment was predicted from the perspective of the nutrient content of crops. In the end, it is also of great significance to further explore the effects of the adjustment of grain planting structure on the consumption of cropland and water resources at the microscale, such as prefecture-level city or county units. In addition, in this paper, the scenario analysis was used to analyze the process of adjustment for grain planting structure in China and its effect on the consumption of cropland and water resources in the past ten years, but it lacks the prediction of grain planting structure and its impact in the future. Next, different scenarios can be set to predict the adjustment of grain planting structure, such as policy intervention, natural disasters and international market environment. Based on this, it is of great significance to explore the impact on the consumption of cropland and water resources for agricultural sustainable development.

Ensuring agricultural production and food security is a systematic project that involves not only natural factors such as water, heat and land, but also social and economic factors such as economic development, urbanization and the international market environment. Therefore, in other words, a sustainable agricultural system is one which is environmentally sound, nonexploitative and which contributes to economic development and social progress [39]. For the stable operation of the agricultural system, it is necessary to encourage the southern regions with rich-water resources to increase the planting area of grain crops and meet its self-sufficiency in grain demand and guide the agricultural areas with water shortage to plant crops in the same season of rain and heat, to improve the spatial matching relationship between the planting of grain crops and cropland and water resources consumption. It is also very important to balance the supply and demand of domestic agricultural products for ensuring food security through international grain market cooperation.

## 5. Conclusions

In this paper, a mathematical econometric model was established by using a scenario comparison analysis method to analyze the process of grain planting structural adjustment and its impact on the consumption of cropland and water resources from 2003 to 2019. The main conclusions were drawn as follows: 

During the study period, China’s outputs of major farm products exhibited a marked upward trend, which can be divided into the rapid growth stage (2003–2015) and the fluctuating growth stage (2016–2019). Meanwhile, the grain planting structure has changed greatly and rice was replaced by corn to become the grain crop with the maximum proportion of planting area since 2007. The increase of corn planting structure proportion was concentrated in the northern regions, such as Northeast China Plain, Northern arid and semiarid region and Loess Plateau.

At the national level, the saving of cropland and water resources consumption showed a “cumulative effect” by the adjustment of the grain planting structure. CRC_l_ and CRCw were 4.62% and 5.72%, respectively. The structural adjustment of grain planting still has a certain effect on saving the consumption of cropland and water resources. From the regional perspective, the impacts of structural adjustment in the northern regions, where cropland resources are relatively abundant but water resources are scarce, are ”saving cropland and water”, such as Northeast China Plain and Northern arid and semiarid region. Meanwhile, that in the southern regions, where water resources are relatively abundant, but cropland resources are scarce, are “saving cropland but not the water”, such as Southern China and Middle-lower Yangtze Plain.

## Figures and Tables

**Figure 1 ijerph-18-07352-f001:**
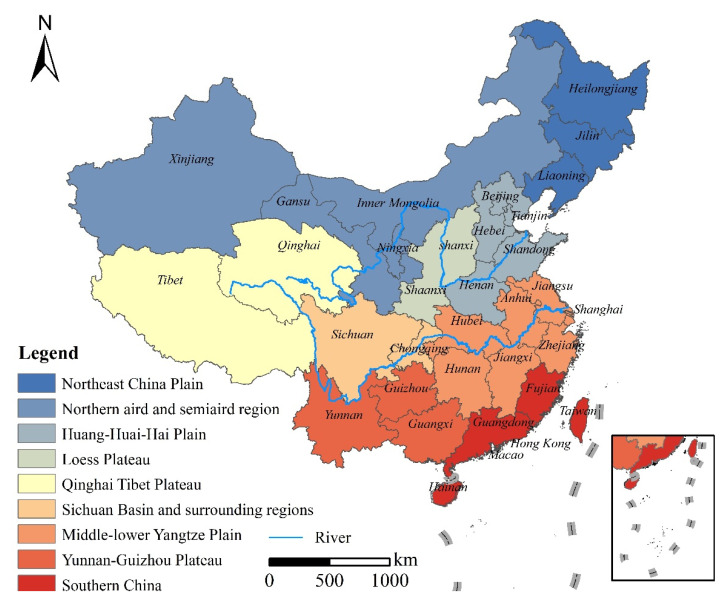
Map of agricultural regions in China.

**Figure 2 ijerph-18-07352-f002:**
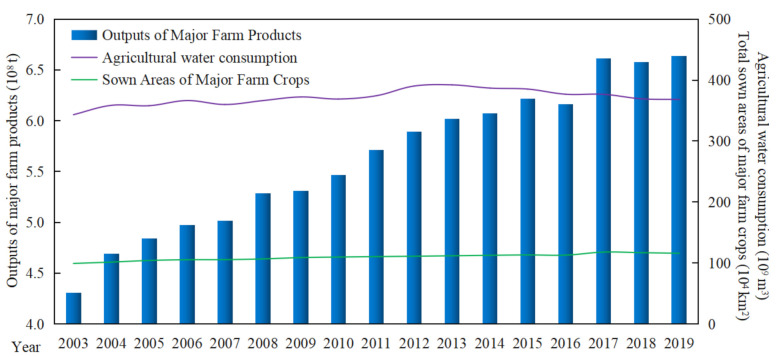
Changes of grain yield and cropland and water resources consumption in China from 2003 to 2019. Sown area of major farm crops (rice, wheat, corn, beans and tubers) and its outputs data were obtained from the China Statistical Yearbook (2004–2020). Agricultural water consumption data were obtained from the China Water Resources Bulletin (2004–2020).

**Figure 3 ijerph-18-07352-f003:**
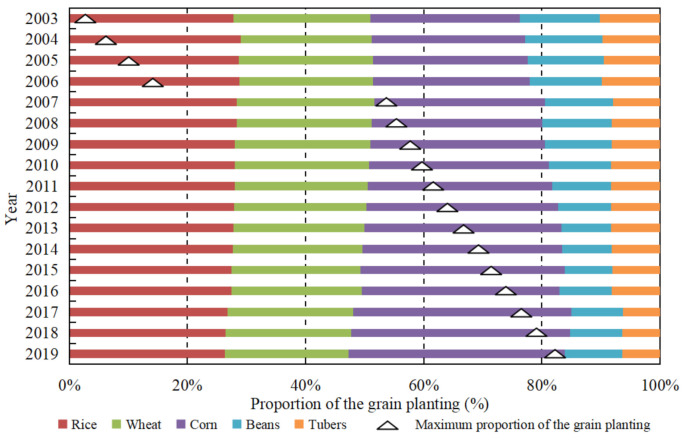
Changes of grain planting structure in China from 2003 to 2019.

**Figure 4 ijerph-18-07352-f004:**
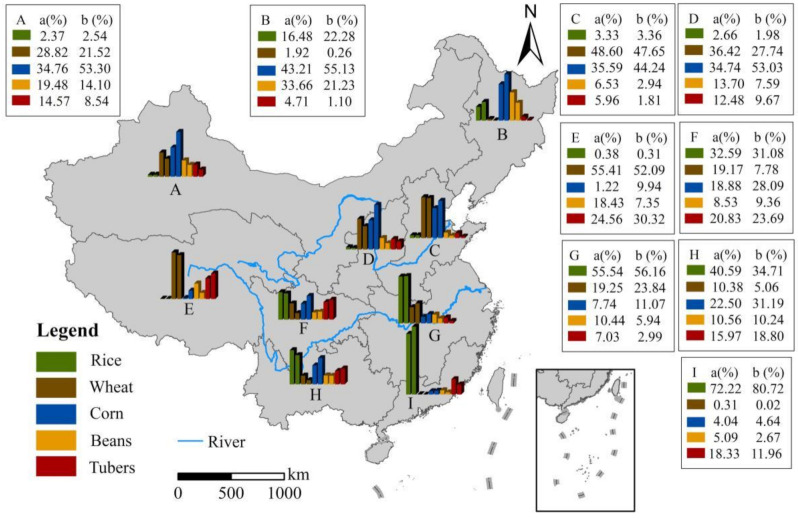
Changes of grain planting structure of China’s nine agricultural regions in 2003 and 2019. (**A**–**I**) represents each agricultural regions, (**A**,**B**) refer to 2003 and 2019, respectively. The values are the planting structure of different food crops.

**Figure 5 ijerph-18-07352-f005:**
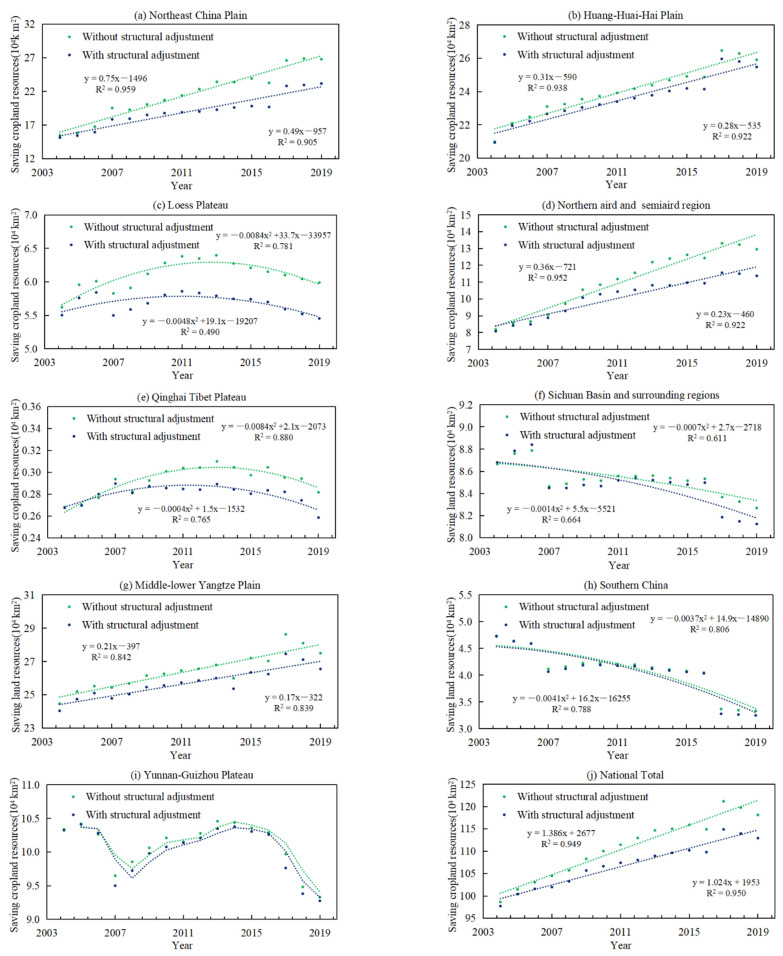
The evolution trend of cropland resources consumption with or without the adjustment of grain planting structure. (**a**,**b**,**d**,**g**,**j**) fit the linear approximation, (**c**,**e**,**f**,**h**) are described by polynomials, (**i**) uses the moving average method to express the evolution trend. y represents the cropland resources consumption with or without the adjustment of grain planting structure, x represents the year, R^2^ expresses the explanatory degree of the relationship between the x and y variables. The values of R^2^ range from 0 to 1. The higher the value, the closer relationship between the x and y variables. Green and blue line are fitted curve, which means the evolution trend of cropland resources consumption without structural adjustment or with the structural adjustment, respectively.

**Figure 6 ijerph-18-07352-f006:**
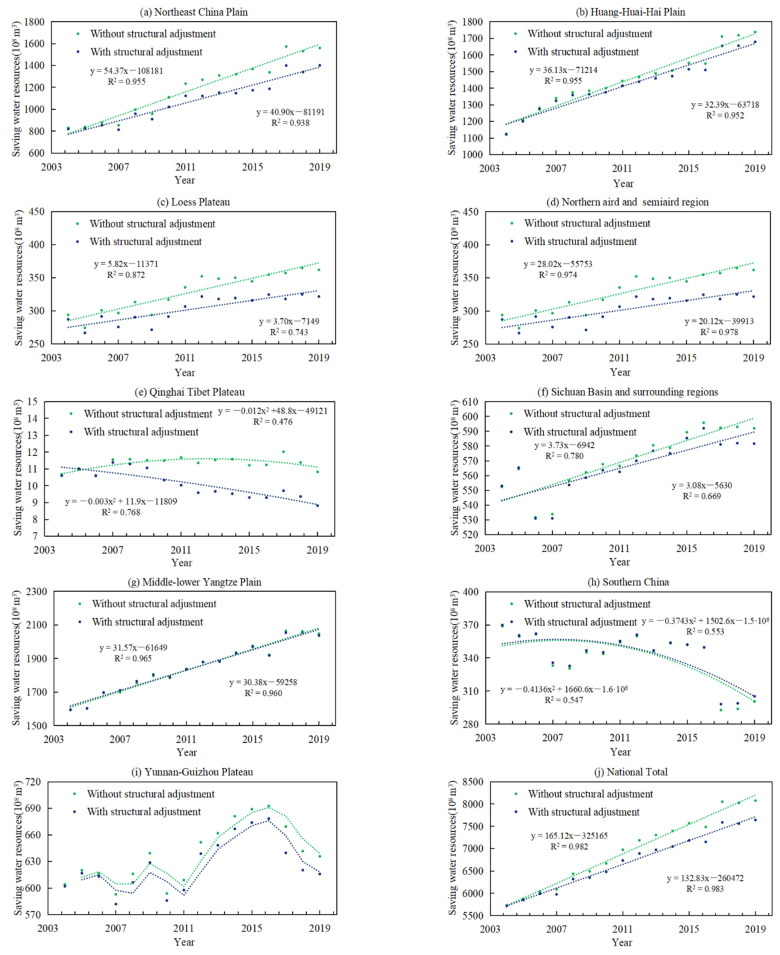
The evolution trend of water resource consumption with or without the adjustment of grain planting structure. (**a**–**d**,**f**,**g**,**j**) fit the linear approximation, (**e**,**h**) are described by polynomials, (**i**) uses the moving average method to express the evolution trend. y represents the water resources consumption with or without the adjustment of grain planting structure, x represents the year, R^2^ expresses the explanatory degree of the relationship between the x and y variables. The values of R^2^ range from 0 to 1. The higher the value, the closer relationship between the x and y variables. Green and blue line are fitted curve, which means the evolution trend of water resources consumption without structural adjustment or with the structural adjustment, respectively.

**Table 1 ijerph-18-07352-t001:** Effect of grain planting structure adjustment on cropland resources consumption during 2003–2019.

Regions	Agricultural Division	Actual Area Changes (10^4^ km^2^)	Area Saved (10^4^ km^2^)	CRC_l_ (%)
The Northern Regions	Northeast China Plain	9.01	3.62	15.61
	Huang-Huai-Hai Plain	4.45	0.43	1.70
	Loess Plateau	0.09	0.54	9.83
	Northern arid and semiarid region	3.53	1.58	13.93
The Southern Regions	Qinghai Tibet Plateau	−0.01	0.02	8.97
	Sichuan Basin and surrounding regions	−0.44	0.15	1.79
	Middle-lower Yangtze Plain	3.51	0.95	3.56
	Southern China	−1.51	0.08	2.57
	Yunnan-Guizhou Plateau	−0.88	0.06	0.62
National Total	-	17.75	5.21	4.62

CRC_l_ means consumption reduction contribution of cropland resources.

**Table 2 ijerph-18-07352-t002:** Effect of grain planting structure adjustment on water resources consumption during 2003–2019.

Regions	Agricultural Division	Actual Water Changes (10^8^ m^3^)	Water Saved (10^8^ m^3^)	CRCw (%)
The Northern Regions	Northeast China Plain	576.42	159.33	11.37
	Huang-Huai-Hai Plain	268.52	59.21	3.53
	Loess Plateau	−2.66	40.44	12.57
	Northern arid and semiarid region	213.14	102.27	14.68
The Southern Regions	Qinghai Tibet Plateau	−1.53	2.02	22.86
	Sichuan Basin and surrounding regions	−31.37	10.33	1.78
	Middle-lower Yangtze Plain	320.05	11.53	0.57
	Southern China	−123.96	−4.62	-1.51
	Yunnan-Guizhou Plateau	−75.65	19.82	3.22
National Total	-	1133.05	437.09	5.72

CRCw—means consumption reduction contribution of water resources.

## Data Availability

Data available in a publicly accessible repository.

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
