# Peer review of "The Adjustment of China’s Grain Planting Structure Reduced the Consumption of Cropland and Water Resources"

_ijerph, 2021, doi:10.3390/ijerph18147352_

Round 1

Reviewer 1 Report

This paper analyzes the process of grain planting structural adjustment and its impact on the consumption of cropland and water resources from 2003 to 2019. The Authors, rather obviously, conclude that switching to the "more performing" corn there will be a saving of cropland and water while increasing the production. Therefore, this paper doesn't add too much to what is rather intuitive. I have some doubts about Figure 5 and Figure 6: I don't think it would be the case to try to linearize the "(i) Yunnan-Guizhou Plateau", I deem that R2=0.137 (Figure 5) and R2=0.295 (Figure 6) are quite ridiculous. I think it would be sufficient to say that blue dots are always below green ones.

As far as the Discussion, I find it bad written, particularly lines 292-293 and lines 301-305 don't make any sense.

Author Response

Dear reviewer,

Thank you very much for the comments and advice. They are really constructive and helpful for us to improve the manuscript.

According to the comments, we have revised our manuscript. The main modification and the responses to the reviewer’s comments are listed as follows:

comment (1): This paper analyzes the process of grain planting structural adjustment and its impact on the consumption of cropland and water resources from 2003 to 2019. The Authors, rather obviously, conclude that switching to the "more performing" corn there will be a saving of cropland and water while increasing the production. Therefore, this paper doesn't add too much to what is rather intuitive. I have some doubts about Figure 5 and Figure 6: I don't think it would be the case to try to linearize the "(i) Yunnan-Guizhou Plateau", I deem that R2=0.137 (Figure 5) and R2=0.295 (Figure 6) are quite ridiculous. I think it would be sufficient to say that blue dots are always below green ones.

#Response: Thank you for your careful review. As you mentioned, the results would be sufficient to say that blue dots are always below green ones, but it's not a linear approximation. We tried to use a reasonable mathematical description. Figure 5, (a)、(b)、(d)、(g) and (j) fit the linear approximation,(c)、(e)、(f) and (h) are described by polynomials, (i) uses the moving average method to express the evolution trend. Figure 6, (a)、(b)、(c)、(d)、(f)、(g) and (j) fit the linear approximation, (e) and (h) are described by polynomials, (i) uses the moving average method to express the evolution trend. Then, we redrew the nonlinear graph in the revised manuscript.

comment (2): As far as the Discussion, I find it bad written, particularly lines 292-293 and lines 301-305 don't make any sense.

#Response: Thank you for your careful review and constructive suggestions. We improved the discussion section. We delete lines 292-293 and lines 301-305, and reword our expression. Revisions in the text are shown using the red highlight for additions. In the discussion section, we mainly expressed three views, 1) As the population grows and wealth increases, the human demand for food is still increasing. Affected by a variety of factors, Global food security faces even more severe challenges. However, China has ensured food security and reduced the consumption of cropland and water resources through the adjustment of grain structure. 2) the consumption quantity of animal food increases year by year, especially the consumption of milk and dairy products and grain consumption decreases significantly. It is possible to increase grain production and reduce the pressure of cropland and water resource consumption by increasing the planting area of high-yielding crops. 3) The spatial movement of grain production barycenter and the spatial dynamic change of grain planting structure aggravated the spatial imbalance of cropland and water resources to some extent. What should be paid attention to is the spatial dislocation between the expansion of grain production and the natural resources, such as cropland, heat and water resources.

In addition, in this paper, we gave a discussion for research methods, line 433-440 in the revised manuscript.

Reviewer 2 Report

The authors proposed a manuscript on the effects of adjustement of grain planting structure on the consumption of croplands and water in China. The paper is well-prepared and below I list only few minor comments and suggestions for improvement:

  1. In the introduction section, please highlight the novelty of this particular study.
  2. Description to Fig. 1: please change to e.g. "Map of agricultural regions in China" (or just remove "nine").
  3. Some results do not fit to linear appoximation. Would it be reasonable to use a different mathematical descirpition?
  4. Please put few sentences (Conclusions section or Discussion) on the future approach to such analyses, e.g. what approach of analysis is good, what should be changed.
  5. Minor errors: L41 ("these" from small letter), L70 ("while" from small letter), L91 ("The basic map... is taken from..."), L301 ("While China is..."), L357 ("...it is necessary..." - it from small letter).

Author Response

Dear  reviewer,

Thank you very much for the comments and advice. They are really constructive and helpful for us to improve the manuscript.

According to the comments, we have revised our manuscript. The main modification and the responses to the comments are listed as follows:

Comment(1): In the introduction section, please highlight the novelty of this particular study.

#Response: Thanks very much for taking the time to review this manuscript. In the introduction section, we added highlight the novelty of this particular study. Through literature review, we found that few studies take care of the effects of grain production on cropland and water resources from the perspective of planting structure. From the perspective of structural adjustment rather than the usual productive factor input, we analyze the process of adjustment for grain planting structure in China and its effect on the consumption of cropland and water resources.

Comment(2): Description to Fig. 1: please change to e.g. "Map of agricultural regions in China" (or just remove "nine").

#Response: Thank you for your careful review. We remved “nine” and Fig. 1 changed to "Map of agricultural regions in China".

Comment(3): Some results do not fit to linear approximation. Would it be reasonable to use a different mathematical description?

#Response: As the reviewer said, some results do not fit a linear approximation. We tried to use a reasonable mathematical description. Figure 5, (a)、(b)、(d)、(g) and (j) fit the linear approximation,(c)、(e)、(f) and (h) are described by polynomials, (i) uses the moving average method to express the evolution trend. Figure 6, (a)、(b)、(c)、(d)、(f)、(g) and (j) fit the linear approximation, (e) and (h) are described by polynomials, (i) uses the moving average method to express the evolution trend. we redrew the nonlinear graph in the revised manuscript.

Comment(4): Please put few sentences (Conclusions section or Discussion) on the future approach to such analyses, e.g. what approach of analysis is good, what should be changed.

#Response:  We have added discussion on the future approach of agricultural structure and food security in the discussion section. Add the following contents:

“In addition, in this paper, the scenario analysis was used to analyze the process of adjustment for grain planting structure in China and its effect on the consumption of cropland and water resources in the past ten years, but it lacks the prediction of grain planting structure and its impact in the future. Next, different scenarios can be set to predict the adjustment of grain planting structure, such as policy intervention, natural disasters, international market environment, etc. Based on this, it is of great significance to explore the impact on the consumption of cropland and water resources for agricultural sustainable development”. Line 433-440 in the revised manuscript.

Comment(5): Minor errors: L41 ("these" from small letter), L70 ("while" from small letter), L91 ("The basic map... is taken from..."), L301 ("While China is..."), L357 ("...it is necessary..." - it from small letter).

#Response: Thank you for your careful review. We are very sorry for the mistakes in this manuscript and the inconvenience they caused in your reading. We corrected the errors in small letters.

Reviewer 3 Report

Here are my comments:

The introduction should be improved, and the authors should follow the guidelines of the authors.
L74: Why authors use the future when they talk about their investigation? It should be in the past or sometimes in the present!
Material and methods section was not well written. The inforamtion for the description and methodology should be expelained in details. 
Also, which program the authors used for the statistical analysis.
Figure 2, the authors have to cite the source of those data! From where they got the data.
L297-298>> Who said this? Please cite this reference for L 297-298 and where it is possible within the MS.
Seleiman, M. F., Selim, S., Alhammad, B. A., Alharbi, B. M., & Cezar Juliatti, F. (2020). Will novel coronavirus (Covid-19) pandemic impact agriculture, food security and animal sectors?. Bioscience Journal , 36(4). https://doi.org/10.14393/BJ-v36n4a2020-54560

and

https://doi.org/10.1080/03066150.2020.1823838

Conclusion is long, authors should make it shorter and focus on the most important findings.

Author Response

Dear  reviewer,

Thank you very much for the comments and advice. They are really constructive and helpful for us to improve the manuscript.

According to the comments, we have revised our manuscript. The main modification and the responses to the comments are listed as follows:

Comment(1): The introduction should be improved, and the authors should follow the guidelines of the authors.

#Response: We read the guidelines of the authors carefully again to improve the introduction section. Such as, we briefly describe the research background of this paper and explain the importance and purpose of this paper based on the literature review. Revisions in the text are shown using the red highlight for additions.

Comment(2): L74: Why authors use the future when they talk about their investigation? It should be in the past or sometimes in the present!

#Response: Thank you for your careful review. We are very sorry for the mistakes in this manuscript. We correct the tense of the sentence as you pointed out.

Comment(3): Material and methods section was not well written. The information for the description and methodology should be explained in detail. Also, which program the authors used for the statistical analysis.

#Response: we added a more detailed interpretation regarding the meaning of the calculation formula. The contents are as follows, if , it means that the adjustment of grain planting structure can help save cropland resources, while if , it means that the adjustment of grain planting structure increases the consumption of cropland resources, and if , it means there is no effect.

In addition, we added the interpretation of the statistical analysis. The last required a bit of explanation that the study results were calculated by Excel and visualized by ArcGIS and Origin software.

Comment(4): Figure 2, the authors have to cite the source of those data! From where they got the data.

#Response: In Figure 2, we added the source of the data. All of the study data is public data obtained from different public publication. Specifically, sown area of major farm crops (rice, wheat, corn, beans and tubers) and its outputs data were obtained from the China Statistical Yearbook (2004-2020). Agricultural water consumption data were obtained from the China Water Resources Bulletin (2004-2020).

Comment(5): L297-298>> Who said this? Please cite this reference for L 297-298 and where it is possible within the MS. Seleiman, M. F., Selim, S., Alhammad, B. A., Alharbi, B. M., & Cezar Juliatti, F. (2020). Will novel coronavirus (Covid-19) pandemic impact agriculture, food security and animal sectors?. Bioscience Journal, 36(4). https://doi.org/10.14393/BJ-v36n4a2020-54560 and https://doi.org/10.1080/03066150.2020.1823838.

#Response: These two important pieces of research found that coronavirus (COVID-19) has adverse effects on agriculture, food security, integrated pest management (IPM), animal productivity, as well, it could lead to disruptions to global food supply chains. We cited these two articles in the discussion.

Besides, in the discussion section, we reword our expression, 1) As the population grows and wealth increases, the human demand for food is still increasing. Affected by a variety of factors, Global food security faces even more severe challenges. However, China has ensured food security and reduced the consumption of cropland and water resources through the adjustment of grain structure. 2) the consumption quantity of animal food increases year by year, especially the consumption of milk and dairy products and grain consumption decreases significantly. It is possible to increase grain production and reduce the pressure of cropland and water resource consumption by increasing the planting area of high-yielding crops. 3) The spatial movement of grain production barycenter and the spatial dynamic change of grain planting structure aggravated the spatial imbalance of cropland and water resources to some extent. What should be paid attention to is the spatial dislocation between the expansion of grain production and the natural resources, such as cropland, heat and water resources.

Comment(6): Conclusion is long, authors should make it shorter and focus on the most important findings.

#Response: We shortened the conclusion section to focus on the most important findings. The first paragraph summarizes the evolution process of China’s grain planting structure from 2003 to 2019. The second paragraph summarizes the effects of the adjustment of grain planting structure on the consumption of cropland and water resources at the national and regional levels.